# Factors Associated with Aging in Place among Community-Dwelling Older Adults in Korea: Findings from a National Survey

**DOI:** 10.3390/ijerph20032740

**Published:** 2023-02-03

**Authors:** Myong Sun Cho, Mi Young Kwon

**Affiliations:** 1Department of Nursing, Gangneung-Wonju National University, 7 Heungeop-myeon, Wonju-si 26403, Republic of Korea; 2Department of Nursing, Ewha Womans University, 52, Ewhayeodae-gil, Seodaemun-gu, Seoul 03760, Republic of Korea

**Keywords:** aging in place, health behavior, older adults, depression, Korea

## Abstract

Ever since baby boomers started turning 65 years old in 2020, Korea is set to become a super-aged society by 2025. This makes it the world’s fastest-aging society. Aging in place (AIP) has become a policy direction to prepare for an aging society and improve older adults’ quality of life. It refers to the ability of older adults to remain in their homes and communities as they age, allowing them to reside in their place of preference and access the services they require to promote their quality of life. A cross-sectional study design was employed using data sampled from the 2020 National Survey of Older Koreans. In total, 9930 older adults (aged between 65 and 99 years old) participated. The results confirmed that the intention to pursue AIP is related to personal factors (education, income, house ownership, smoking, exercise, depression), interpersonal and communal factors (unmet healthcare needs, need for home care services, family contact), and policy level factors (basic pension beneficiary, long-term care services) using an ecological model. The findings may promote individual health behaviors and help fill the gap between unmet healthcare needs and community care services that positively influence older adults’ AIP.

## 1. Introduction

In 2021, it was projected that the population aged 65 years and over would account for 16.5% of Korea’s population. Korea is projected to become a super-aged society by 2025, when the older adult population will reach 20.6% of the total [1,2]. Globally, older adults have a higher prevalence of health problems such as back pain, osteoarthritis, chronic obstructive pulmonary disease, diabetes, depression, and dementia, and these health conditions are the factors that affect their well-being and life satisfaction the most [3].

In 2020, the average number of chronic diseases per older adult aged over 65 years in Korea was 1.9, and 54.9% were patients with comorbidities and two or more chronic diseases. The prevalence rate by disease type was high blood pressure (56.8%), followed by diabetes (24.2%), hyperlipidemia (17.1%), and osteoarthritis or rheumatoid arthritis (16.5%) [4].

Due to the health problems experienced by older adults, Korea’s health insurance medical expenses for those aged 65 years and older increased from 10.49 trillion Korean won in 2008 to 21.36 trillion Korean won in 2015 and 40.61 trillion Korean won in 2021, an increase of 8.4% from the previous year. As such, 43.4% of Korea’s total medical expenses are dedicated towards older adults, and it is expected to increase to 65.4% by 2030, increasing the burden on national medical expenses [5].

The World Health Organization (WHO) proposed Age-Friendly Cities and Communities (AFCC) to encourage local communities to consider and solve problems together from a preventive and positive perspective so that all countries worldwide can naturally accept “aging” and older adults can lead active lives [6]. The Global Network of Age-friendly Cities covers eight interconnected sectors, namely community and healthcare, transportation, housing, social participation, outdoor spaces and buildings, respect and social inclusion, civic participation and employment, and communication and information [6].

Aging in place (AIP) refers to an approach in which older adults continue to live in familiar homes and communities by receiving appropriate support and services, despite their disabilities or limitations [7]. Promoting older adults’ ability to participate and contribute to their communities and society, providing integrative and primary care services tailored to individual needs, and providing access to long-term care for older adults in need are key. In 2020, the period from 2021 to 2030 was declared the “Decade of Healthy Aging” at the United Nations General Assembly [7].

AIP policies are supported by extensive literature demonstrating older adults’ preference to stay in their homes as they age. It is a universal direction even in countries that experienced aging before Korea [8].

Due to the increasing older adult population and the burden of medical expenses, a policy shift has occurred in the direction of allowing older adults to independently receive necessary services and continue living in the community rather than in facilities or hospitals. Therefore, in terms of cost, continuing to live in the same place is evaluated as more cost-effective than care services in hospitals or older adult care facilities [9].

As such, the policy for AIP must meet several conditions to be realized. In other words, various services such as housing, medical care, and daily care should be provided appropriately so that older adults can live at home safely and these services should respond to their changing needs [10]. Continuity of care must also be ensured such that various services are provided through integration between professionals and care systems [11].

Since 2018, the Korean government has been promoting an Integrated Community Care policy that provides housing, medical care, nursing care, and daily care services in local communities to build a community environment for AIP [12]. In a 2020 survey on older adults, 88.6% of them responded that they plan to continue living at home [4], and the Korean government is seeking policies to strengthen community preventive services to support AIP and improve medical access for older adults [12].

For community-dwelling older adults, preventive services such as healthcare, dementia prevention, and emotional support are provided in an integrated way according to their functional and health status so that they can maintain a healthy life in old age without going to a hospital or facility.

In previous studies, the factors affecting the continued residence of older adults in the community were identified as follows: demographic factors including sex [13], age [14,15,16], education level [14], financial status [17], and housing ownership and satisfaction [13,18,19]; physical and psychological health-related factors including independent activities of daily living (IADL) [11,20], chronic disease and depression [21], health behaviors such as smoking and exercise [22], and self-esteem [23]; and social and environmental factors including social contact with family, friends, and neighbors [21], community satisfaction [17], unmet medical needs [19], safety and security, and residential satisfaction [24]. Moreover, basic pensions and non-contributory pensions have shown positive effects on mental health and reduced psychosocial stress associated with financial hardship. It also increased the likelihood of independent living [25].

As such, AIP, which is an important strategy to ensure a healthy and satisfying life for older adults, has a complex effect on various dimensions such as health behavior, physical and psychological health status, and living environment in addition to demographic factors of older adults [26,27].

So far, studies on continued residence in local communities in Korea have mainly been conducted in one region and promote continued residence in the community. These studies have focused on the analysis of needs [28] and environmental factors affecting attitudes toward continued residence [13,17,29]. Recent research has found a stronger association between physical and psychological health and AIP [30]. Current studies on AIP in Korea have studied various types of housing welfare services centered on economic dependence and dependence in daily life in terms of housing and welfare, gerontology (definition and application of AIP in the Korean context), housing, urban planning, welfare needs, and satisfaction, among others.

In particular, in terms of healthcare, very few studies examine the AIP intentions and related factors of older adults. Although several studies have investigated older adults’ preferences for AIP, little is known about health-related factors [30]. Therefore, there is a limit to comprehensively identifying factors affecting AIP.

The purpose of this study is to identify the AIP intentions of older adults aged 65 years and older living in the local community using data from a survey on older adults in Korea and to analyze the factors influencing these intentions using an ecological approach, which explains how different environmental system types affect human development [31]. Based on these results, we aim to provide suggestions for the direction of health and welfare services and policies for older adults.

## 2. Materials and Methods

This study used raw data from the nationwide survey of older adults aged 65 years and over called the “2020 National Survey of Older Koreans”, conducted in 2020 by the Korea Institute for Health and Social Affairs. To sample the original data, the sample size of the two-stage probability proportional to size systematic sampling target was 10,000 people aged 65 years or older living in their own homes in 17 subnational jurisdictions, such as special cities, metropolitan cities, and provinces nationwide in the 2018 census survey [4].

The survey was conducted with a final sample comprising 10,097 older adults who agreed to participate in 2020 (https://data.kihasa.re.kr/kihasa/kor/databank/DatabankDetail.html) (accessed on 29 July 2021). The questionnaire was completed using computer-assisted personal interviews conducted by trained interviewers. The study was conducted according to the guidelines of the Declaration of Helsinki and approved by the Institutional Review Board of the Korea Institute for Health and Social Affairs (No. 2020-36, 8 July 2020).

The survey questionnaire consisted of nine areas, namely economic status, family and social relations, exchange of help between families, health status and health behavior, functional status and nursing care, cognitive function, economic activity, leisure and social activity, and living environment [4]. The 10,097 participants over the age of 65 years were asked “Where would you like to live if you remain in good health?” Finally, weight was applied to the data of 9930 participants, excluding 167 who did not respond to the question, and the data were analyzed.

Therefore, in this study, based on the ecological framework [32], we examined the factors affecting AIP at the intrapersonal, interpersonal and community, and policy levels (Figure 1).

The intrapersonal level refers to biological and sociodemographic characteristics.They include sex; age; education; income; health-risk behaviors; and mental, physical, and functional health.The interpersonal and community level refers to an individual’s social network, consisting of interactions with family and medical and social service systems.They include region, community satisfaction, exposure to crime, unmet needs for healthcare services, community care service needs, and social contact.The policy level refers to economic, social, and political policies and regulations including pension and long-term care.

### 2.1. Measurements

#### 2.1.1. Dependent Variable

AIP intention was used to determine older adults’ plans to continue living in the community [24]. Under Korea’s policy direction, AIP aims to provide housing, nursing care, welfare, and medical services so that people can age in the place where they live. Previous studies have measured AIP as a single question about AIP intention [24]; in this study, it was measured as follows.

The question “Where would you like to live if you remain healthy?” was asked with the possible responses being “Continue to live in the current house”, “Plan to move to a house with a better living environment”, and “Plan to move to a house where meals and living convenience services are provided”. Therefore, in this study, in a narrow sense, AIP intention was defined as “Continuing to live in the current house”. If participants indicated that they would prefer to keep living in their homes, they were categorized as the “AIP group” and the others as the “Not AIP group”.

Although there is controversy about the reliability of variable measurement using a single item, participants could accurately and clearly recognize the intention of the questionnaire; therefore, a measurement containing a single, global question is likely to suffice [33].

#### 2.1.2. Independent Variable

Based on the literature review, intrapersonal level variables were included and measured, such as sociodemographic factors (sex [13], age [14,15,16], educational level [14], household income [17]), psychological factors (depressive symptoms), health-risk behaviors (cigarette smoking, exercise) [21,22], physical health factors (Korean version of Instrumental Activities of Daily Living [K-IADL]), cognitive impairment, number of comorbidities), house ownership, and residential satisfaction [18,24].

Residential satisfaction, community satisfaction, and a number of comorbidities were measured as continuous variables. Age (65–74 years, 75 years and older), sex (male, female), educational level (primary, middle, high school), annual household income in quartiles (494, 495–899, 900–1999, 2000 Korean won), living arrangements (living alone, living together), house ownership (no, yes), current smoker (no, yes), physical exercise (no, yes), depressive symptoms (no, yes), fall experience (no, yes), and cognitive impairment (no, yes) were measured as categorical variables.

Cigarette smoking and physical exercise were assessed using the following two questions: “Do you smoke cigarettes currently?” and “Do you exercise currently?” The possible responses were “No” and “Yes”.

Depressive symptoms were assessed using the Korean version of the Geriatric Depression Scale-Short Form (SGDS-K), which was originally developed in English and translated into Korean [34]. The SGDS-K comprises 15 items with response options of “Yes = 1” or “No = 0”. In this study, a score ≥ 8 was taken as indicative of significant depressive symptoms. The SGDS-K has satisfactory validity and reliability [35] (Cronbach’s alpha was 0.73). The number of comorbidities as a continuous variable was counted against a specific list of diseases. The list was compiled based on the 2017 National Survey of Older Koreans [7] and modified by the authors to include certain additional diseases.

The number of comorbidities was measured using the question “Have you ever been diagnosed as having one of these diseases for more than 3 months?” Comorbidity status was assessed among 29 chronic diseases such as hypertension, cerebrovascular disease, chronic pulmonary disease, congestive heart failure, dementia, and diabetes [28].

The K-IADL scale was employed to assess physical dependency. Participants were asked whether they needed assistance in ten different instrumental activities such as personal hygiene and grooming, housekeeping, and preparing meals [36]. K-IADL scores were categorized as having no limitations in daily activities (K-IADL = 0) and having limitations in daily activities (K-IADL ≥ 3 or 4). K-IADL scores ranged from 0–33 points, with higher scores indicating greater dependence on instrumental activities of daily living. Cronbach’s alpha value was 0.94 in the validity-identifying study [36], and in this study, it was 0.96.

Fall experience was assessed using the question “Have you had a fall in the past year?” The response options were “No” and “Yes”.

Cognitive impairment was assessed using the Mini-Mental State Examination (MMSE), which is the most widely used screening instrument [37]. The Korean version of the MMSE contains 19 items, and the maximum score is 30 points. The Korean MMSE showed satisfactory validity and reliability (Cronbach’s alpha was 0.81) in this study (Cronbach’s alpha was 0.90).

Interpersonal and community-level variables were included, such as region (urban, rural), community satisfaction [18,24], crime exposure (no, yes), unmet needs for healthcare services (no, yes), frequency of kin contact [24] (none, less than once per month, more than once per month), and score for community care service needs. Community satisfaction and score for community care service needs were measured as continuous variables.

Residential satisfaction was assessed with the following question using a 5-point Likert scale (very unsatisfied, 1; very satisfied, 5) measuring, “How satisfied are you with the residence that you currently live in?”.

Crime exposure was assessed using the following question: “Have you experienced any of the following six types of crimes or accidents in the past year: Property crime, violence, fraudulent purchase experience, phone fraud, pedestrian traffic accident experience, fire safety accident?”.

Unmet healthcare needs were assessed using the following question: “In the past year, have you thought that you needed treatment, but you did not receive treatment at a hospital?”.

Scores for community satisfaction were assessed using a 5-point Likert scale (very unsatisfied, 1; very satisfied, 5) for seven types of community facilities and amenities such as social welfare and medical facilities, public transportation, green space, public security, distance from the family residence, opportunities to interact with neighbors, and overall community environment. The questions had satisfactory validity and reliability in this study (Cronbach’s alpha was 0.83), and the score for community satisfaction was used as a continuous variable summing up each response.

Frequency of kin contact was assessed with the following question: “In the past year, how often have you been in contact with family members who live apart from you (interaction via phone, mobile phone text message, social networking service, e-mail, letter, etc.)”. Possible responses were “Almost every day”, “About once a week”, “About 1–2 times a month”, “About 2–3 times a week”, “About 1–2 times every three months”, and “About 1–2 times a year”. Responses were re-categorized as none, less than once per month, and more than once per month based on the results of a previous study on intergenerational contacts; the most frequent contact was more than once a month in the case of Korean older adults [38].

Two types of services are officially available to older adults who have difficulty living independently in Korea. This includes long-term care and community care services. Community care services are provided to vulnerable older adults who are not eligible for long-term care. It is difficult to receive overlapping benefits as these are provided in a complementary manner.

The long-term care insurance benefits are assistance with excretion, bathing, eating, cooking, washing, nursing, and treatment or recuperation counseling. Community care services include safety and safety confirmation, social relationship improvement programs, physical and mental health education and counselling, transport support, and housework support, which are temporarily provided to vulnerable elderly people who cannot receive long-term care benefits.

Hence, considering the content and context of community care services, we have classified them as interpersonal and community interactions.

Community care service needs were assessed with a six-part question using a 5-point Likert scale (very unsatisfied, 1; very satisfied, 5): “To what extent do you think older adults need the following six types of community care services for AIP: residential environment improvement, daily life support, safety support (regular safety checks, rescue support in case of emergency, etc.), transport to health facilities, home visiting by health professionals, and provision of information and counseling?” Scores for community care service needs were used as a continuous variable summing up each response.

Policy level variables were included, such as basic pension and the number of long-term care services used. The basic pension scheme program provided monthly benefits that mitigate the situation for older persons aged 65+ who live in poverty, particularly for individuals not covered by the compulsory national pension in Korea. The eligibility for the targeted pension benefit is determined based only on the applicant’s age, income, and property [39]. We measured several long-term care services used. The number of long-term care services was divided into six categories of home care services, including day/night care center services, home-visit services to promote cognition activities, home-visit nursing services, home-visit bathing services, short-term institutionalized care, and provision of welfare devices [40].

### 2.2. Statistical Analysis

The chi-square test was used to assess the difference between the AIP group and the Not AIP group with independent variables. Multivariable logistic regression models were used to assess the factors affecting AIP intention. Model 2 used interpersonal and community-level factors. Model 3 had policy-level factors, and after controlling for Model 1, the intrapersonal-level factors were entered.

Statistical significance in this study was defined as a *p*-value < 0.05, and adjusted *p*-values with odds ratios (ORs) and 95% confidence intervals (CIs) were calculated using SPSS Statistics 27.0 (IBM Corp, Armonk, NY, USA). To ensure unbiased national estimates, sampling weights were computed for participants to ensure that the sample was representative of all adults aged 65 years and older in Korea.

## 3. Results

### 3.1. General Characteristics of the Study Population

Of the 9930 community-dwelling older adults who participated in the survey, 43.1% were men and 56.9% were women, aged 73.68 ± 6.46 years. The mean age of the AIP group (73.79 years) was greater than that of the Not AIP group (73.10 years). Of the total, 83.8% of the participants preferred to stay in their homes. Regarding educational level, 58.2% of the participants had graduated from middle school or higher, while 80% were living together; 79.8% were homeowners; 11.9% were smokers; 8.3% had depressive symptoms; and 7.1%, 3.9%, and 2.5% experienced falls, crime, and unmet healthcare needs, respectively.

The two groups, AIP and Not AIP, showed differences for several variables. Examples include age, educational level, income level, living arrangements, house ownership, smoking, physical exercise, depressive symptoms, falls and crime experience, contact with kin, and pension (Table 1).

### 3.2. Differences in the Level of Community Care Service Needs

The average score of community care service needs in order from highest to lowest was as follows: counseling services (providing necessary information for daily life, etc.), improvement of living conditions such as home repairs, safety support (regular safety checks, rescue service linking hospital support in case of emergency, etc.), and transportation support services to a hospital. The average score difference between the AIP and Not AIP groups was statistically significant for all of the aforementioned services except for daily life support and visiting medical and health services (Table 2).

### 3.3. Factors Affecting AIP

Logistic regression analysis was conducted after controlling for sociodemographic factors such as sex, age, education, income, and living arrangements.

Model 1 included intrapersonal-level factors such as sex, age, education, income, and health-related factors. In Model 2, we added interpersonal and community-level factors. In Model 3, we added the policy-level factors that we are interested in. Newly added variables showed a significant improvement in R^2^ and were 0.196, (*p* < 0.001), 0.217 (*p* < 0.001), and 0.221 (*p* < 0.001).

Participants who were 75 years and older were more likely to age in place than those aged between 65–74 years. Those with higher education levels and higher income were also more likely to age in place, as were those who owned a house or reported a higher score for residential satisfaction. Those with depressive symptoms, smokers, and those with a fall experience were less likely to age in place. Regardless of the frequency of exercise, those who engaged in physical exercise were more likely to age in place than those who did not.

Those who had been exposed to crime had unmet healthcare service needs and greater needs for residential services and were less likely to age in place than those who had not been exposed to crime. Moreover, those who lived in rural areas were more likely to age in place than those who lived in urban areas. Those who reported having family contact less than once per month were more likely to age in place than those who had no family contact. Lastly, those who were covered by a basic pension were more likely to age in place than those who were not (Table 3).

## 4. Discussion

### 4.1. Basic Pension and Long-Term Care Service

The basic pension, which was introduced to support the stability of low-income seniors and promote their welfare, had a positive effect on AIP intention. In previous studies [41,42,43], receipt of a basic pension was found to increase AIP intention. Under the basic pension scheme, the key is to reduce poverty in older adults. It covers 66.7% of the individuals aged 65 years and older who were earning below the ceiling, which was KRW 1,800,000, for a single person and KRW 2,880,000 for a couple [44]. The eligibility for the basic pension benefit is determined based only on the applicant’s age, income, and property. The basic pension is complementary to the public pension; it is a contributory pension, and it reduces financial stress and increases the likelihood of AIP in older adults.

The percentage of individual public pension income differs between the young-old adult and old-old adult groups. Young-old adults receive 40% of the public pension income, 75–79-year-olds receive 30.4%, 80–84-year-olds receive 19.7%, and 85+ year-olds receive 9.3% since older people received less money [4]. For healthy aging and AIP, it is necessary to protect older adults from poverty through social security systems. To maximize the functional capacity of older adults and maintain their autonomy and dignity, a broader system is needed to ensure that long-term care insurance functions as a basic safety net [7].

The number of long-term care services used was not significant to AIP intention. However, Korean older adults would still prefer to receive long-term care in their own homes instead of moving into a nursing home or senior living facility. Long-term care service appears to be a necessary but not sufficient condition for AIP. In other words, to achieve AIP, the establishment of a long-term care service provision system must be implemented, and follow-up studies are needed to identify the relationship between the two.

### 4.2. Age

In this study, the old–old adults (aged 75 years or older) were more likely to intend to age in place than the young–old adults (65 to 74 years), which is consistent with previous studies [15,16]. The characteristics of older adults by age were related to education level, economic level, and social contact. In general, the young–old adults had higher education levels, higher economic levels, and more active social networks. In addition, it was found that for the young–old adults, the decision to move was made for reasons such as housing convenience and the high accessibility of local convenience facilities [45].

In addition, the main reason for the old–old adults to refrain from migration and decide to age in place is due to a decrease in physical ability and increased dependence [46]. As age increases, there is a tendency to experience physical difficulties, and informal networks appear to weaken. Functional limitation rates for those over 85 years old were 67.4% and fall rates were 22.0% [4]. In the case of health level, it tended to deteriorate with age, and in particular, the functional state appeared to deteriorate rapidly around the age of 80.

However, regarding the relationship between age and AIP, a study found that AIP intention increased with age [41,47], showing inconsistent results. The young–old adults—the 65-year-old baby boomers born right after the Korean War—are a generation that has experienced historical events and changes ranging from industrialization, urbanization, and democratization to the recent informatization in Korea [4]. Thus, they are healthier, wealthier, more active, and more confident than the old–old adults as they benefited from social changes such as the expansion of public education, economic development, development of the social security system, and internationalization through the development of social and economic systems. In conclusion, it is inferred that various characteristics such as education level, wealth, and values were reflected in AIP even between the young–old and old–old adult groups.

### 4.3. Health-Related Factors

Smoking decreased AIP while physical activity increased it. Maintaining healthy behaviors, such as regular physical activity and smoking cessation, is known to reduce the risk of non-communicable diseases among adults, improve physical and mental abilities, and reduce care dependence [3].

As described above, the healthier the lifestyle, the higher the likelihood of AIP intention [30,43]. Psychological and emotional health conditions such as depression, anxiety, and dissatisfaction due to aging appear more frequently in those living in care facilities than in those continuing to live in their own residences. It is understood that older adults may be more inclined to move to a combined facility than live in the current house [48].

Since the experience of falling significantly lowers AIP and is a major cause of the burden of disease in older adults, it is necessary to maintain a safe environment. In addition, if there is exposure to crime or a fall incident, the intention to age in place is significantly lowered, so it is necessary to apply a safe and barrier-free design to houses where older adults live and improve the residential environment. As most older adults are living in their own houses, it is necessary to install appropriate facilities and equipment (e.g., threshold removal, bathroom handles, safety calls, anti-slip mats, etc.).

### 4.4. Unmet Healthcare Needs and Community Care Service Needs

In this study, it was found that unmet needs for healthcare reduced AIP intention. The management of complex chronic diseases or geriatric syndromes must reshape the healthcare system to meet the needs of the older adult population, with a focus on maintaining capacity as people age [7].

Reflecting on the research results to facilitate AIP intention, society needs to provide older adults with a broad range of community-based assistance to meet their needs such as affordable and accessible housing, access to health and support services, and social connections.

In this study, it was found that the higher the unmet needs for healthcare and the community care needs score, the lower the AIP. This is because the relationship between healthcare and community care is complementary [49].

Transportation support services, emergency referral services, and the provision of necessary information mentioned in Table 2 can facilitate decision-making for visits to medical institutions and healthcare services utilization. In this case, the mere existence of a caregiver improves access to medical care and reduces preventable medical use [11]. Care services serve as substitutes for or are complementary to health services. For example, when care is not provided, older adults’ health may deteriorate and diseases may occur (increasing medical needs). However, when care is provided, older adults’ health status improves and their medical needs decrease. For example, system-wide efforts and formal policies to ensure older adult-centered services in Canada were found to reduce institutional costs and home care visits [50].

The WHO recommends delivering care through a community-based workforce, supported by community-based services. The composition of the workforce that provides such care depends on the environment and resources available. It is made possible through the mixing and matching of health and social welfare professionals such as doctors, nurses, related health professionals, social workers, community health workers, and volunteers such as family members [51].

### 4.5. House Ownership and Residential Satisfaction

In this study, 79.8% of the participants owned a house, and owning a house increased AIP intention. In the same vein, it was found that home ownership (OR = 0.23) and residential satisfaction (OR = 0.6) lowered moving plans [52]. In previous studies, older adults who owned a house were more likely to stay in their house than those who rented a house [15,41,47].

Considering the decrease in income, home ownership is directly related to the stability of life in old age. However, older adults who do not own a home are more likely to move to nursing facilities and nursing homes where food, clothing, and shelter can be provided simultaneously [52].

In the older adult group, the proportion of residents living in houses that had been built more than 30 years ago was 32.8%, which was higher than 16.5% of the general population, and residential satisfaction was lower in the older adult group (2.93%) than in the general population (2.97%). Therefore, the older adult group’s need for housing improvement/renovation in-kind and financial loan support was higher (21.8%) than in the general population [52]. As seen above, the older adult group lives in older houses compared to the general population and therefore, has a high dissatisfaction level due to noise and uncomfortable home structures, which are understood to lower AIP intention.

To improve residential satisfaction, it is necessary to remove or improve the risk factors (high threshold, stairs, toilet, bathroom, etc.) [53]. It is also necessary to consider providing mobility devices including canes, walkers, and wheelchairs to assist older adults with declining physical functions. In other words, the improvement of the residential environment so that older adults can live safely improves their quality of life including their health status; this is supported by research results [54].

Specifically, to increase residential satisfaction, there are requirements such as housing repairs, assistance in securing housing, support for maintaining a rental contract, counseling related to remodeling and repair, and resource connection. The WHO recommends intergenerational interventions that bring together the younger generation and older adults to work on a common project, such as co-housing, with supporting policies and education for understanding and reducing ageism [55].

### 4.6. Social Network

In the results of this study, it was found that contact with kin less than once a month increased the likelihood of AIP intention, but kin contact more than once a month was not statistically significant for AIP intention.

Kinship generally provides older adults with more stable and normative contact and support over the course of their lives. Intimate kin relationships provide major social support to older adults, and the quality of support given and received in these relationships affects health. In addition, the non-kinship relationships that older adults form during their lifetime provide support that is different from that provided by kinship relationships. As a result, these various social network configurations lower anxiety, stress, and depression [33].

Older adults form kin and non-kin relationships during their life trajectories, and these relationships provide different structures and functions to older adults. The results of this study showed that a certain level of kinship affects AIP intention, and a comprehensive study on the frequency, duration, and intensity of kinship and those lacking kinship is needed in future studies on the AIP intention effect of social networks.

## 5. Conclusions

The purpose of this study was to identify the AIP intentions of older adults living in Korea. Our study is meaningful as it empirically and comprehensively identified the factors affecting AIP using a nationwide representative sample of older adults. Based on the results, the study suggests the direction of health and welfare services and policies for AIP as follows:

First, this study highlights the significance of health-related factors such as healthy behaviors, mental health, and healthcare service provision in increasing AIP. Thus, it is necessary to introduce a policy and coordinated mechanism that lays the foundation for linking necessary healthcare and welfare services in an efficient way. Through the use of these services, older adults can maintain independent function despite the decline in physical function that appears according to the stage of aging.

Second, house ownership and residential satisfaction strongly affect AIP intention. Prior to the introduction of senior housing, which requires substantial time and budget to prepare and build, at the current stage, remodeling the housing to meet older adult needs requires the support of local governments and communities and the provision of orthosis and equipment to prevent falls in older adults.

Despite the meaningful results of this study, there are some limitations. First, among the various definitions of AIP intention, considering them realistically, there are not many intermediate options between houses and facilities in Korean society; in this study, the questions are structured based on the operational definition of “continue to live in the current house” in a narrow sense. However, AIP intention has a broader sense, from aging at their own home to growing old in a living environment chosen by the person with a social connection to their community regardless of whether it is their current house or another residence chosen by them. In future studies, it is necessary to develop and apply a measurement tool with high reliability and validity while measuring AIP intention in the Korean context.

Second, since the participants used a self-report questionnaire, it is possible that they responded based on desirability rather than their actual memories and experiences, which may have affected the variables investigated. Third, since the data used in this study were cross-sectional rather than longitudinal, there is generally no evidence of a temporal sequence between intrapersonal, interpersonal, community, and policy-level factors and AIP.

AIP, which is emphasized as an important strategy to ensure a healthy and satisfying life for older adults, reflects the trajectory of life, including health behaviors, physical and psychological health status, and living environment, in addition to the demographic and sociological factors of older adults. Therefore, it is designed and provided in a way that can provide healthcare and community care through a person-centered mix while reflecting these various dimensions.

AIP intention can be strengthened through the integration of housing and social care that combine services that consider the needs and desires of older adults and the improvement of the underdeveloped residential environment in which older adults currently reside. This can prevent unnecessary relocation to nursing facilities and hospitals and reduce the burden of social care for older adults.

## Figures and Tables

**Figure 1 ijerph-20-02740-f001:**
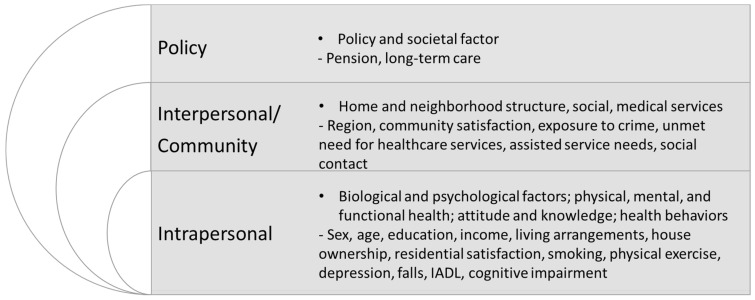
Conceptual framework and factors associated with aging in place.

**Table 1 ijerph-20-02740-t001:** General characteristics of Not AIP and AIP groups (n = 9930).

Factor		N (Weighted %)			t or f or χ^2^	*p*-Value
		Total (n = 9930)	Not AIP (n = 1605)	AIP (n = 8326)		
Sex	Male	4282(43.1)	671(41.8)	3611(43.4)	1.30	0.255
	Female	5648(56.9)	933(58.2)	4715(56.6)		
Age	65–74 years	5656(57.0)	1004(62.6)	4652(55.9)	24.78	<0.001
	≥75 years	4274(43.0)	600(37.4)	3674(44.1)		
Educational level	Primary	4157(41.9)	561(35.0)	3596(43.2)	37.88	<0.001
	Middle	2322(23.4)	430(26.8)	1892(22.7)		
	High	3451(34.8)	613(38.2)	2838(34.1)		
Annual household income quartiles (10,000 Korean won)	0–494	2466(24.8)	431(26.9)	2035(24.4)	23.66	<0.001
	495–899	2481(25.0)	331(20.6)	2150(25.8)		
	900–1999	2499(25.2)	397(24.7)	2102(25.2)		
	above 2000	2486(25.0)	446(27.8)	2040(24.5)		
	Living together	7945(80.0)	1188(74.1)	6757(81.2)	42.28	<0.001
Living arrangements	Living alone	1985(20.0)	416(25.9)	1569(18.8)		
House ownership	No	2008(20.2)	718(44.8)	1290(15.5)	714.20	<0.001
	Yes	7922(79.8)	886(55.2)	7036(84.5)		
Current smoker	No	8747(88.1)	1361(84.9)	7386(88.7)	19.09	<0.001
	Yes	1183(11.9)	243(15.1)	940(11.3)		
Physical exercise	No	4559(45.9)	793(49.4)	3766(45.2)	9.45	0.002
	Yes	5372(54.1)	812(50.6)	4560(54.8)		
Depressive symptoms	No	9105(91.7)	1422(88.7)	7683(92.3)	23.18	<0.001
	Yes	825(8.3)	182(11.3)	643(7.7)		
Fall experience	No	9222(92.9)	1429(89.1)	7793(93.6)	41.29	<0.001
	Yes	708(7.1)	175(10.9)	533(6.4)		
Cognitive impairment	No	6733(67.8)	1108(69.1)	5625(67.6)	1.42	0.234
	Yes	3197(32.2)	496(30.9)	2701(32.4)		
Region	Urban	7518(75.7)	1309(81.6)	6209(74.6)	36.19	<0.001
	Rural	2412(24.3)	295(18.4)	2117(25.4)		
Crime exposure	No	9545(96.1)	1467(91.4)	8078(97)	113.75	<0.001
	Yes	386(3.9)	138(8.6)	248(3)		
Unmet healthcare needs	No	9684(97.5)	1541(96.1)	8143(97.8)	16.66	<0.001
	Yes	246(2.5)	63(3.9)	183(2.2)		
Frequency of kin contact	None	597(6.0)	155(9.7)	442(5.3)	60.75	<0.001
	<1/month	891(9.0)	97(6.0)	794(9.5)		
	≥1/month	8444(85.0)	1353(84.3)	7091(85.2)		
Basic pensioner	No	2844(28.6)	543(33.8)	2301(27.6)	25.27	<0.001
	Yes	7087(71.4)	1062(66.2)	6025(72.4)		
Annual income		1658.96 ± 55.21	1552.32 ± 28.62	1569.55 ± 25.60	1.53	0.125
Age (years)		73.68 ± 6.46	73.10 ± 6.3	73.79 ± 6.48	−3.92	<0.001
Residential satisfaction		3.82 ± 0.66	3.43 ± 0.75	3.89 ± 0.61	−22.79	<0.001
Number of comorbidities		1.9 ± 1.53	2.01 ± 1.61	1.87 ± 1.51	3.29	0.001
K-IADL ^1^		10.63 ± 2.56	10.73 ± 2.73	10.61 ± 2.52	1.58	0.113
Community satisfaction		25.87 ± 3.83	25.37 ± 3.96	25.96 ± 3.80	−5.54	<0.001
The total score of community care service needs		22.29 ± 4.68	22.78 ± 4.73	22.20 ± 4.66	4.55	<0.001
Number of long-term care services		0.03 ± 0.29	0.03 ± 0.27	0.03 ± 0.30	−0.26	0.794

^1^ K-IADL = Korean-instrumental activities of daily living.

**Table 2 ijerph-20-02740-t002:** Differences in the level of community care service needs.

	Total	Not AIP	AIP	*t*	*p*-Value
Improvement of living conditions such as home repairs	2.35 ± 0.87	2.18 ± 0.82	2.38 ± 0.88	−9.09	<0.001
Daily life support (housework, nursing, bathing, meal support, etc.)	2.27 ± 0.99	2.23 ± 1.05	2.28 ± 0.98	−1.72	0.086
Safety support (regular safety checks, emergency referrals, etc.)	2.24 ± 0.98	2.15 ± 0.98	2.26 ± 0.97	−4.27	<0.001
Visiting medical and health services by doctors and nurses	2.17 ± 0.98	2.15 ± 0.98	2.18 ± 0.98	−1.04	0.298
Transportation support services to a hospital	2.24 ± 0.95	2.16 ± 0.98	2.25 ± 0.95	−3.56	<0.001
Counseling services (providing necessary information for daily life, etc.)	2.44 ± 0.88	2.36 ± 0.92	2.45 ± 0.88	−3.78	<0.001

*p* < 0.05.

**Table 3 ijerph-20-02740-t003:** Logistic regression analysis of factors associated with AIP.

Factor	Model 1		Model 2		Model 3	
	Odds Ratio (95% CI)	*p*-Value	Odds Ratio (95% CI)	*p*-Value	Odds Ratio (95% CI)	*p*-Value
Female (ref = male)	0.87 (0.76–1.00)	0.049	0.91 (0.79–1.04)	0.162	0.9 (0.79–1.04)	0.150
≥75 years (ref = 65–74 years)	1.30 (1.13–1.49)	<0.001	1.30 (1.13–1.49)	<0.001	1.24 (1.07–1.42)	0.003
Educational level (ref = primary school)						
Middle school	0.63 (0.54–0.74)	<0.001	0.67 (0.57–0.79)	<0.001	0.68 (0.58–0.80)	<0.001
High school	0.56 (0.48–0.66)	<0.001	0.62 (0.52–0.73)	<0.001	0.67 (0.56–0.79)	<0.001
Annual household income quartiles	0.86 (0.81–0.91)	<0.001	0.86 (0.81–0.91)	<0.001	0.88 (0.83–0.93)	<0.001
Living together (ref = living alone)	0.89 (0.77–1.03)	0.116	0.89 (0.76–1.03)	0.107	0.88 (0.76–1.02)	0.094
House ownership (ref = no ownership)	3.62 (3.19–4.12)	<0.001	3.58 (3.14–4.09)	<0.001	3.64 (3.19–4.16)	<0.001
Residential satisfaction	2.46 (2.26–2.69)	<0.001	2.45 (2.23–2.68)	<0.001	2.49 (2.27–2.73)	<0.001
Smoking (ref = no smoking)	0.79 (0.66–0.94)	0.010	0.78 (0.65–0.94)	0.009	0.79 (0.65–0.95)	0.011
Physical exercise (ref = no exercise)	1.12 (0.99–1.26)	0.062	1.12 (0.99–1.26)	0.064	1.13 (1.00–1.28)	0.048
Depressive symptoms (ref = no symptom)	0.65 (0.54–0.80)	<0.001	0.64 (0.5–0.78)	<0.001	0.64 (0.52–0.78)	<0.001
Number of comorbidities	1.00 (0.96–1.04)	0.837	1.01 (0.9–1.05)	0.528	1.01 (0.97–1.05)	0.557
Fall experience (ref = no)	0.59 (0.48–0.72)	<0.001	0.63 (0.52–0.78)	<0.001	0.63 (0.51–0.77)	<0.001
K-IADL	1.02 (0.99–1.04)	0.122	1.02 (1.00–1.05)	0.057	1.02 (0.99–1.04)	0.215
Cognitive impairment (ref = no)	1.08 (0.95–1.24)	0.249	1.11 (0.96–1.27)	0.155	1.09 (0.95–1.26)	0.201
Rural (ref = urban)			1.39 (1.19–1.62)	<0.001	1.36 (1.17–1.58)	<0.001
Community satisfaction			1.01 (1.00–1.03)	0.124	1.01 (1.00–1.03)	0.116
Crime exposure (ref = no)			0.39 (0.30–0.49)	<0.001	0.41 (0.32–0.52)	<0.001
Unmet healthcare service needs (ref = no)			0.68 (0.49–0.94)	0.021	0.66 (0.48–0.92)	0.015
The score for community care service needs			0.97 (0.96–0.99)	<0.001	0.97 (0.96–0.99)	<0.001
Frequency of kin contact (ref = none)						
<1/month			2.34 (1.71–3.19)	<0.001	2.29 (1.68–3.13)	<0.001
≥1/month			1.18 (0.95–1.47)	0.138	1.17 (0.94–1.46)	0.165
Basic pension (ref = no)					1.42 (1.24–1.64)	<0.001
Number of long-term care services					1.19 (0.94–1.52)	0.148
R^2^ (*p*-value)	0.196 (*p* < 0.001)		0.217 (*p* < 0.001)		0.221 (*p* < 0.001)	

*p* < 0.05; K-IADL = Korean-instrumental activities of daily living.

## Data Availability

Data were made available by the Korea Institute for Health and Social Affairs after permission was obtained on 29 July 2021 (https://data.kihasa.re.kr/kihasa/kor/contents/ContentsList.html, accessed on 29 July 2021).

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
