# Peer review of "Factors Associated with Aging in Place among Community-Dwelling Older Adults in Korea: Findings from a National Survey"

_ijerph, 2023, doi:10.3390/ijerph20032740_

Round 1
Reviewer 1 Report
This is an interesting and useful article outlining the overwhelming desire by older adults to continue to age in place, rather than move into residential facilities. Overall, the article reads well, however the authors may wish to consider the following points:
Line 10: Should this read ‘Since baby boomers turned 65 years old in 2020, Korea is set to become …’
Line 27-28: This sentence states that Korea is projected to become a super-ageing society by 2030. However, in the abstract, it says that Korea is due to become a super-aged society by 2025.
Line 29: ‘Globally, older adults have numerous health problems such as ….’. Currently, this sentence reads as though all older adults have all these problems. The authors may wish to consider rewriting this sentence e.g. ‘Older adults have a higher prevalence of health problems such as ....’
Line 44: Please add the full title for organisations such as WHO in the first instance. This also applies to Line 58 - UN.
Line 114: The first part of the materials and methods section currently reads as though this is planned study (e.g., Line 114 ‘The study plans to use….’, Line 135 ‘we intent to …’) rather than a study that has already been conducted.
Line 154 – 160: I note that the authors chose to only include ‘Continue to live in the current house’ as their AIP option. The reviewer queries why ‘Plan to move to a house with a better living environment’ was not considered as an AIP option as well (in the conclusions – Lines 456-459 -you mention that there may be limitations in only choosing the ‘Continue to live in the current house’ option; and the article also notes that there is low housing satisfaction in the older age group).
Line 196 – 198: The first sentence states that no limitations in daily activities is equivalent to a score of 0 on the K-IADL, however the second sentence states that the K-IADL scores range from 10-33.
Lines 209 – 213: The authors may wish to remove the response types for crime exposure, unmet healthcare needs, frequency of kin variables that are then further defined in lines 218-249.
Lines 277 – 283: For clarity, the authors may wish to specify what statistical significance means for these results – e.g., were these variables associated with AIP or not.
Lines 322-323: The reviewer found this sentence difficult to understand. Should this read ‘However, regarding the relationship between age and AIP, this study found that AIP intention increases with age, which is inconsistent with previous research’.
Lines 337 – 340: The authors may wish to consider re-writing these two sentences. For a reader unfamiliar with the Korean pension income, the reviewer was unsure if this means that the public pension decreases as you age, that there are different polices which mean public pension rates differed over time, or something else.
Lines 421 – 423: If the reviewer is reading this correctly, older adults with less kin contact were more likely to AIP. The authors may wish to expand on why they think this is the case (the following paragraphs mention the importance of kin relationships, but could place more emphasis on why these kin relationships were less important in this sample).
Author Response
December 29, 2022
International Journal of Environmental Research and Public Health
Dear reviewers:
Thank you for giving us the opportunity to submit a revised draft of our manuscript titled "Factors Associated with Aging in Place among Community-Dwelling Older Adults in Korea: Findings from a National Survey" to the International Journal of Environmental Research and Public Health. We appreciate the time and effort that you and the reviewers have dedicated to providing valuable feedback on our manuscript. We are grateful to the reviewers for their insightful comments on our paper. We have incorporated changes to reflect most of the suggestions provided by the reviewers and have highlighted the changes within the manuscript.
Here is a point-by-point response to the reviewers’ comments and concerns.

Reviewer 2 Report
Using an “ecological framework”, this paper presented cross-sectional analyses from a national dataset of older adults in Korea to examine the associations of 3 levels of factors and aging-in-place intention. The topic is important and the study seems properly conducted; however, it seems that the data available/used had limited capacity in testing the hypothesized AIP framework in a rigorous manner. Overall, the writing was acceptable; however, proofreading by a professional editor with scientific background would improve its quality.
Below are some comments for the authors to consider to strengthen the study and the manuscript.
MAJOR COMMENTS:
Introduction
Can the authors explain what they meant by an “ecological approach” and provide its reference in the Introduction? The term was mentioned in line 110 in the Introduction with no elaboration, and the reference was presented in the Materials and Methods section (line 135). I thought the most well-known ecological approach in the human development literature would be Bronfenbrenner’s ecological systems theory; however, this was not what the authors referred to. Reference 30 (Lau et al.) referred by the authors appears incorrect.
Materials and Methods
Broadly speaking, the factors in the ecological framework (P.3, lines 135-146), the factors in Figure 1, and the variables in 2.1 Measurements section need better alignment. For instance, economic policy is presented as a policy level factor (line 145) in the framework; however, it is missing at the policy level in Figure 1 and it is not assessed in this study.
Assisted service needs or IADL is often considered an indicator of functional health in the literature, which in this model was considered an intrapersonal level factor. However, assisted service needs was considered an “interpersonal and community level factor” in this model. Can the authors reconcile the discrepancy? Also, in many aging studies, when I-ALD is measured, ADL is measured as well. Was ADL assessed in the 2020 National Survey of Older Koreans? It would be informative to compare ADL and I-ADL as predictors of AIP intention.
2.1 Measurement: Do most/ all community-dwelling older adults in Korea live in houses? Do any live in flats/ apartments, tenements, or boats? I wonder if those who do not live in houses may find it difficult to relate to the focus on houses in the AIP intention item (the outcome variable) and in the assessment of “house satisfaction” (predictor variable) (lines 215- 217). If there are indeed people who live in places that are not houses, would it be more suitable to refer to their homes as residences instead of “houses”, and to ask about “home satisfaction” or “residence satisfaction” instead of “house satisfaction”? Please address these queries, and clarify if this was just a matter of translation, or if most/ all community-dwelling older adults in Korea indeed live in houses.
2.1.1 Dependent variable: The assessment of AIP intention seems problematic. Not only was it measured by a single item, none of its response options clearly indicate moving from a home in the community to an institution (e.g. assisted living facility, nursing home), which I think is the definition of not aging in place, that is, the real “No AIP group”. As such, those who plan to move to an institution would likely find none of the options applicable. It seems the current “AIP group” would be more accurately labelled as “AIP and no relocation plan group”, and the current “No API group” as “AIP with relocation plan group”. Can the authors explain how they selected this dependent variable and the response options, and what they have done to address its limitation?
2.1.2 Independent variable
P.4, line 168: Is pension not considered to be a source of household income among older adults in this sample? Please clarify.
P. 6, line 242 – 250: It seems arbitrary to put “community care service needs” under interpersonal and community level variables and “long term care services” under policy level variables, since there are public policies that deal with community care services and long term care services. Please explain.
P. 6, line 250: Was it the case that only two “policy level variables” were assessed in the 2020 National Survey of Older Koreans and included in the data analyses? If so, please do not present basic pension and number of long term care services as examples (line 251-251). Also, as these two variables were entered in the final step in Model 3 of the logistic regression, it seems as if the authors wished to examine their associations with the outcome variable after adjusting for all of the other variables. Given the assumed importance of these 2 variables in the authors’ model, can they provide more information on how they were defined and measured in this section? For instance, what does it mean to be a “basic pensioner (Table 1)”? What would be counted by participants as long term care services in this study?
Results
3.3 Factors affecting AIP: Can the authors make clear why some factors were entered in Model 1, other set of factors in Model 2, and a third of factors in Model 3? The intention of examining the association of some factors with the outcome variable “after controlling for” the variables already in the model was only clearly presented in the beginning of this section (line 288-289), but not in the rest of this section.
Discussion
Based on the order in which the variables were entered into the logistic regression, it seems that the authors hypothesized pension and long term care services (“policy/ societal” level factors) were the most important predictors. If so, please present these results in the beginning of the Discussion section. Also, as asked before, is pension considered a source of income among older adults in this sample? Please clarify this in the Methods section.
Please expand the limitations based on the revisions made in response to the comments above.
MINOR COMMENTS
- Since this study has been completed, please revise descriptions that suggest the study has yet to be conducted (section 2, lines 114-116, “… plans to…”). Please check the whole manuscript for similar instances and correct as appropriate.
- “Old older adults” may be better called “older olds” or “older old adults”. “Young older adults” may be better called “younger olds”. Please take reference from publications about older age groups in high impact journals, and check the whole manuscript for similar instances and correct as appropriate.
Table 3: Please add "Odds ratio" in the column headings.
- The authors are encouraged to use more precise descriptions in the whole manuscript. Please revise and seek help from a professional editor as needed. Examples:
o “the higher the health status is perceived to be, the higher the AIP” (p.10, line 349). I think this statement referred to the finding about depressive symptoms only, not health status, hence the statement was not an accurate description of the finding. “The higher the AIP” was imprecise as well as AIP group membership was a binary variable.
o When the authors cautioned the readers against inferring causality from cross-sectional and correlational data, it would have been better if they had stated that explicitly instead of using a general statement of “caution is required in interpretation” (p.12, line 465). Please check the whole manuscript for similar instances and correct as appropriate.
Author Response

(The authors gave the same response as above.)
